# Decadal decrease in Los Angeles methane emissions is much smaller than bottom-up estimates

Zhao-Cheng Zeng [1] ✉, Thomas Pongetti[2], Sally Newman [1,8], Tomohiro Oda [3,4,5], Kevin Gurney [6], Paul I. Palmer [7], Yuk L. Yung [1,2] & Stanley P. Sander [1,2] ✉

Methane, a powerful greenhouse gas, has a short atmospheric lifetime (~12 years), so that emissions reductions will have a rapid impact on climate forcing. In megacities such as Los Angeles (LA), natural gas (NG) leakage is the primary atmospheric methane source. The magnitudes and trends of fugitive NG emissions are largely unknown and need to be quantified to verify compliance with emission reduction targets. Here we use atmospheric remote sensing data to show that, in contrast to the observed global increase in methane emissions, LA area emissions decreased during 2011-2020 at a mean rate of $(-1.57 \pm 0.41)$ %/yr. However, the NG utility calculations indicate a much larger negative emissions trend of $-5.8$ %/yr. The large difference between top-down and bottom-up trends reflects the uncertainties in estimating the achieved emissions reductions. Actions taken in LA can be a blueprint for COP28 and future efforts to reduce methane emissions.

Atmospheric methane ($CH_4$) is a potent greenhouse gas (GHG) with about 80 times higher global warming potential than carbon dioxide ($CO_2$) over a 20-year period[1]. Because of its relatively short lifetime (~12 years), reducing emissions of $CH_4$ can have an immediate contribution to slowing global warming[1,2]. Major human activity-related sectors responsible for increasing $CH_4$ globally include emissions from livestock, oil and gas industries, landfills, coal mining, rice paddies, and water treatment plants[3]. Urban regions, such as the Los Angeles (LA) basin and the U.S. East Coast, have been found to be major sources of fugitive $CH_4$ emissions[4,5], probably due to the leaky natural gas infrastructures such as pipelines and end-user appliances[6], suggesting that these regions can be very important targets for cutting $CH_4$ emissions. However, our understanding of $CH_4$ emissions from urban regions is still very limited and underexamined.

Significant $CH_4$ emissions in the LA basin, the second most important urban carbon-emitting region in the US, have been previously reported from top-down estimates using various sources of measurements[4,7–15]. The annual emissions in the LA basin for the past decade are roughly $400 \pm 150$ Gg/year from previous studies (see Summary Figs. in ref. 13,14), which account for about a quarter of total emissions in the state of California (1545 Gg/year in 2016, according to the California Air Resources Board (CARB) inventory). Significant effort has been devoted to determining the relative emissions from fossil (e.g., natural gas supply) and biogenic sources (e.g., landfill emissions). Previous studies[4,12] used the ethane ($C_2H_6$) to $CH_4$ ratio as a tracer of fossil sources and showed that most of the excess $CH_4$ emissions in the basin can be attributed to uncombusted losses from the natural gas system. The conclusion agrees with results from using $CH_4$ isotopologues[16] and mobile measurements of $C_2H_6$ to $CH_4$ ratios[17].

[1]Geological and Planetary Sciences, California Institute of Technology, Pasadena, CA, USA. [2]Jet Propulsion Laboratory, California Institute of Technology, Pasadena, CA, USA. [3]Earth from Space Institute, Universities Space Research Association (USRA), Columbia, MD, USA. [4]Department of Atmospheric and Oceanic Science, University of Maryland, College Park, MD, USA. [5]Graduate School of Engineering, Osaka University, Suita, Osaka, Japan. [6]School of Informatics, Computing, and Cyber Systems, Northern Arizona University, Flagstaff, AZ, USA. [7]School of GeoSciences, University of Edinburgh, Edinburgh, UK. [8]Present address: Planning and Climate Protection Division, Bay Area Air Quality Management District, San Francisco, CA, USA. ✉e-mail: zcz@gps.caltech.edu; stanley.p.sander@jpl.nasa.gov

The California Methane Survey[18] used airborne imaging spectroscopy to find that fugitive $CH_4$ emissions from super-emitters of oil and gas infrastructure contribute significantly to the fugitive emissions.

In an effort to reduce greenhouse gas emissions, legislation in the State of California mandates reductions in $CH_4$ emissions by 40% below 2013 levels by 2030 (ref. [19]). The provisions of California Senate Bill (SB)1371(ref. [20]) was approved on September 21, 2014. SB1371 specifically targets reducing natural gas leakage from the Public Utilities Commission-regulated gas pipeline facilities that are intrastate transmission and distribution lines. To verify compliance with California law it is not sufficient to rely on self-reported, bottom-up emission inventories that contain large uncertainties. Measurement, reporting, and verification (MRV)[21] should be implemented to ensure that the emissions controls are working in the long-term.

In this study, we analyze the decadal trend (2011–2020) of $CH_4$ emissions in the LA basin using measurements from the California Laboratory for Atmospheric Remote Sensing–Fourier Transform Spectrometer (CLARS-FTS), operated by the Jet Propulsion Laboratory (JPL). JPL's CLARS-FTS has been measuring trace gases and inferring emissions continuously since September, 2011, thus providing the longest available data record that covers the entire LA basin (Supplementary Fig. S1). The tracer-tracer ratio method[13] that relates the emissions to mixing ratio enhancement is adopted here. Previous studies have used CLARS-FTS data to investigate the seasonal cycle of methane emissions in LA[11] and to infer the natural gas leakage rate using data from 2011 to 2017 (ref. [15]). Here, we focus on the decadal trend of $CH_4$ emissions and updates the inferred leakage rates of the natural gas system for the past decade in the basin. With an improved understanding of the long-term trend of urban $CH_4$ emissions in Los Angeles, we will have more insight into the effectiveness of future control measures and mitigation strategies to reduce GHG emissions in cities.

## Results and discussion

### Seasonal cycles of $CH_4$ emissions in LA

The seasonal variability of $CH_4$ emissions in the LA basin was first reported[11] using CLARS-FTS data and confirmed based on in-situ measurements from the Los Angeles Megacity Carbon Project[14]. Before estimating emissions, we calculated the seasonal cycle of the excess ratio ($XCH_{4,xs}/XCO_{2,xs}$), which is an indicator of the $CH_4$ emissions relative to the $CO_2$ emissions. Figure 1a shows consistent and significant seasonal cycles year by year, with peak values in winter and minimum values in summer. Interestingly, a similar seasonal cycle of excess ratio can be inferred from National Oceanic and Atmospheric Administration/Mount Wilson Observatory (NOAA/MWO) flask data (Supplementary Text 1), which measures the ambient air with significant contribution from the up-slope flow from the basin. These MWO flask data have been found to be sensitive to the anthropogenic emissions in the LA basin[8]. Since $CO_2$ emissions in the basin do not have a large seasonality according to bottom-up inventories (Supplementary Fig. S2), the seasonality in $XCH_{4,xs}/XCO_{2,xs}$ is primarily driven by the seasonality of $CH_4$ emissions. This seasonality has also been reported for Boston[22] and Washington, D.C[23]. Note that from March through May 2020, the excess ratio is mostly higher than in the previous year, resulting mainly from the sharp decrease in $CO_2$ emissions due to the COVID-19 pandemic lockdown[24]. After determining the seasonal cycle of the excess ratio, we then estimate the monthly $CH_4$ emissions based on $CO_2$ emissions of ODIAC and CARB, respectively, using the tracer-tracer inversion method (see Methods). However, the biogenic $CO_2$ fluxes in LA show significant seasonality[25,26] and need to be taken into account when converting $XCO_{2,xs}$ to $XCO_{2,ff}$ by removing the biogenic contribution, where $XCO_{2,ff}$ represents the excess resulting from fossil fuel emissions only. We adopt the conversion factor ($CO_{2,ff}/CO_{2,xs}$) from multi-year (2006–2016) isotope measurements by Newman et al.[25]. We also compare our results with those

derived from the data collected in 2015 by Miller et al.[26] (see Methods). The results of estimated monthly $CH_4$ emissions are shown in Fig. 1b. The seasonality is generally consistent with the excess ratio. The dip in $CH_4$ emissions in April 2020 is uncertain and needs further confirmation. However, we note that the estimated $CH_4$ emissions at Boston University (BU) in Boston[22] also showed a decrease in April 2020. The marked decrease in methane emissions at BU may be due to reduced appliance use in office buildings, restaurants, and/or the BU campus surrounding the BU site, or other beyond-the-meter losses[22].

### Estimation of the leakage rate of natural gas systems

The significant correlation between the natural gas consumed in the basin and $CH_4$ emissions to the atmosphere was first reported based on CLARS-FTS observations[15]. Here we extended the data through 2020 and found that the variabilities in natural gas consumption from the residential and commercial sectors can explain about half of the variations in $CH_4$ emissions ($R^2 = 0.55$), as shown in Fig. 2a. The non-seasonal component, determined by the y-intercept in Fig. 2b, is $10.56 \pm 1.80$ Gg $CH_4$/month. We obtain similar statistics from $CH_4$ emissions based on the CARB inventory ($R^2 = 0.48$; intercept $= 11.27 \pm 1.81$ Gg $CH_4$/month), as shown in Fig. 2c. The mean intercept and slope are $10.92 \pm 1.28$ Gg $CH_4$/month and $2.8 \pm 0.18\%$, respectively. The significant correlation may be explained by the fugitive methane emissions from natural gas systems in the basin with a static leakage rate. The fugitive emissions may be attributed to the natural gas infrastructure such as distribution pipelines or the many post-meter leaks (e.g., from home appliances) that can accumulate to give large emissions[6]. Based on this assumption, if the seasonal correlation is causal, about $(2.8 \pm 0.18)\%$ of the commercial and residential natural gas consumption in LA is released into the atmosphere, according to the regression slope as shown in Fig. 2b. This is comparable within uncertainty to the $(2.5 \pm 0.5)\%$ loss rate of natural gas in Boston based on in-situ measurements from 2012–2020 (ref. [22]). Both sets of estimates are at the lower bounds of the estimates by Wennberg et al.[4], which showed a loss rate of approximately 2.5–6% of the natural gas delivered to basin customers. As a comparison, the correlation of NG usage- and $CH_4$ emissions based on the biogenic fluxes in Miller et al.[26], shows a larger slope of $3.6 \pm 0.21\%$ (Supplementary Fig. S9).

### Decadal trend of $CH_4$ emissions in LA

A statistical model consisting of a linear component and a seasonal component comprised of harmonic functions (see Methods) is fitted to the monthly $CH_4$ emissions from 2011 to 2020. The fitting results, as shown in Fig. 3a, b, show significant decreasing trends of $-0.35 \pm 0.20$ Gg/month, which is $(-1.05 \pm 0.59)\%$/yr relative to the mean monthly emission of 33.74 Gg/month, based on the ODIAC inventory, and $-0.68 \pm 0.19$ Gg/month, which is $(-2.08 \pm 0.58)\%$/yr relative to the mean monthly emission of 32.87 Gg/month, based on the CARB inventory. The average of the decreasing trend is $(-1.57 \pm 0.41)\%$/yr. The most likely explanation is that for the past decade, because of California legislation mandating $CH_4$ emissions reductions, efforts have been made to identify sources and cut emissions. This steady decreasing trend, therefore, demonstrates the effectiveness of $CH_4$ emission control measures in the LA basin.

To further understand the possible drivers of the interannual trend, ensemble empirical mode decomposition (EEMD) analysis was carried out to determine the interannual trend from the derived methane emission time series and relate the trend to changes in policies. EEMD is a powerful tool for extracting trend information from nonlinear and nonstationary time series (see Methods). The interannual trend of $CH_4$ emissions extracted from the EEMD analysis, as shown in Fig. 3c, shows a large drop starting around 2015. This inflection point in emissions occurs around the years when the provisions of SB1371 (approved on September 21, 2014) came into effect. With the approval of this bill, it is reasonable to assume that the most

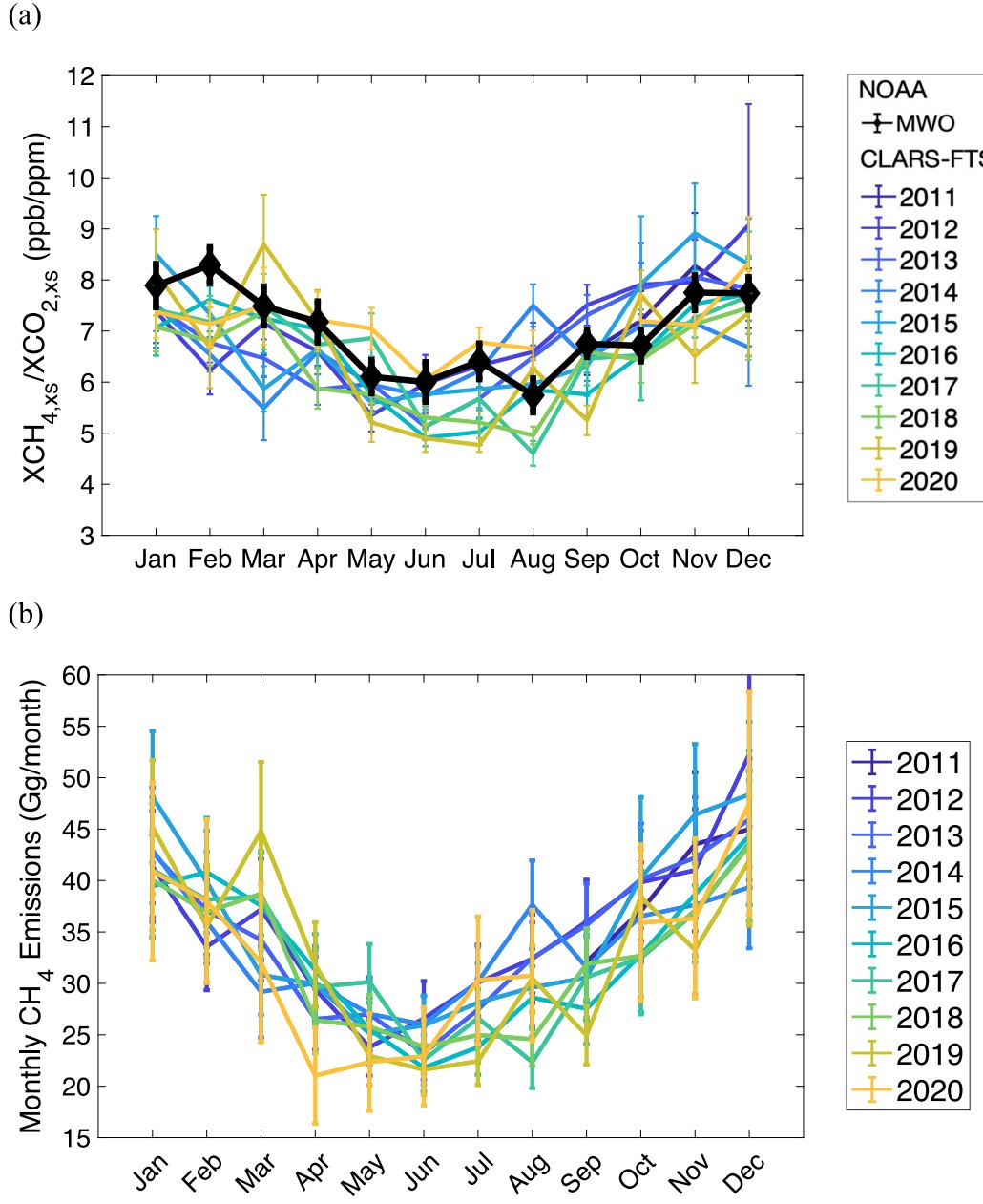

**Fig. 1 | Seasonal cycles of excess ratio and CH₄ emissions. a** Monthly ratio of excess XCH₄ to excess XCO₂, i.e. XCH₄,xs/XCO₂,xs, from 2011 to 2020 calculated from CLARS-FTS observations. Data for September 2020 are not available due to instrument shutdown during wildfires. As a comparison, the corresponding excess ratios derived using NOAA's Mt. Wilson Observatory (MWO) flask measurements averaged over 2011–2020 are shown in black; (**b**) Monthly CH₄ emissions in the LA basin from 2011 to 2020 are estimated based on the CO₂ emissions from ODIAC and the derived monthly XCH₄,xs/XCO₂,ff excess ratio after correcting for the biogenic flux contribution. The error bars represent the estimation uncertainty (1σ) of the monthly values.

rapid progress would have been made in the first few years after 2015 because the NG utility likely targeted the largest leaks first.

To investigate the spatial patterns of the excess ratio, we further examine three subregions in the LA basin: western, central, and eastern. However, the results show no significant spatial differences (Supplementary Fig. S13) perhaps because the spatial distribution of fugitive emissions across the basin is relatively uniform. In addition, for distant reflection points, CLARS-FTS integrates across a relatively long optical path in the basin which complicates the identification of individual point sources. In contrast, the integrated basin emissions are robust.

The goal of reducing emissions of short-lived climate pollutants by 2030 relative to the 2013 level can be achieved by capturing or avoiding methane emissions from a variety of sources including dairy manure, enteric fermentation, disposal of organics at landfills, and fugitive methane emissions[27]. The observed decrease in CH₄ emissions inferred from CLARS-FTS measurements demonstrates the effectiveness of California legislation beginning with AB 32, the Global Warming Solutions Act, in 2006. From Supplementary Fig. S11, we also see an interesting decreasing interannual trend of emissions that occurs in the second half of the year which drives the decreasing trend of emissions for the past decade. The reason for this decrease is not clear yet and will need further study. In the LA basin, however, several previous studies[4,12,16,17] have shown that most of the CH₄ emissions to the atmosphere come from fugitive emissions. This suggests that future emissions reduction efforts should focus on natural gas infrastructure and end-use.

(a)

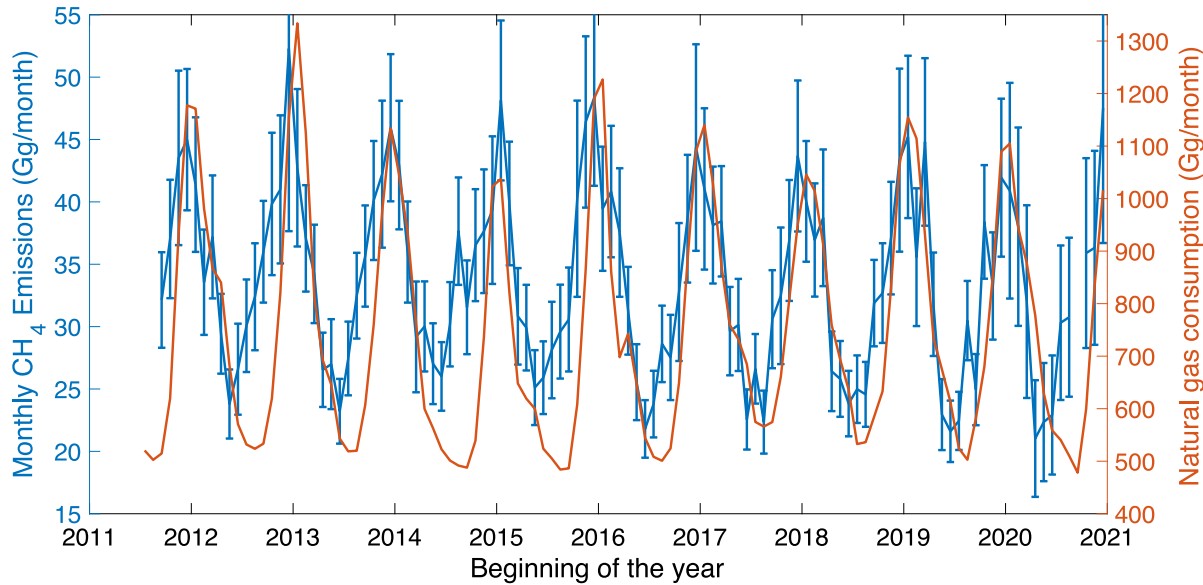

(b)

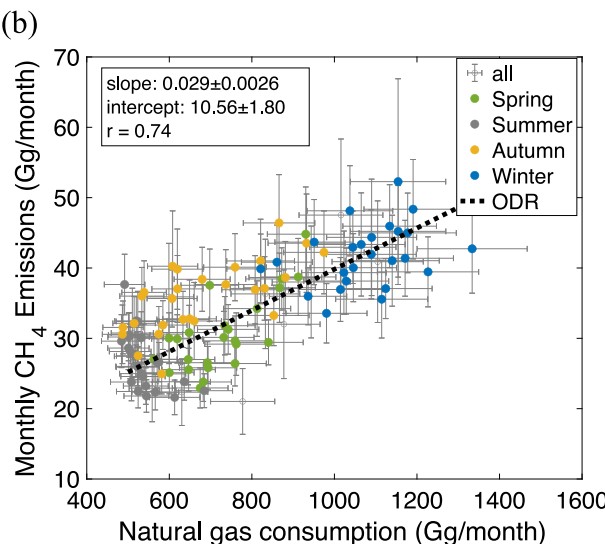

(c)

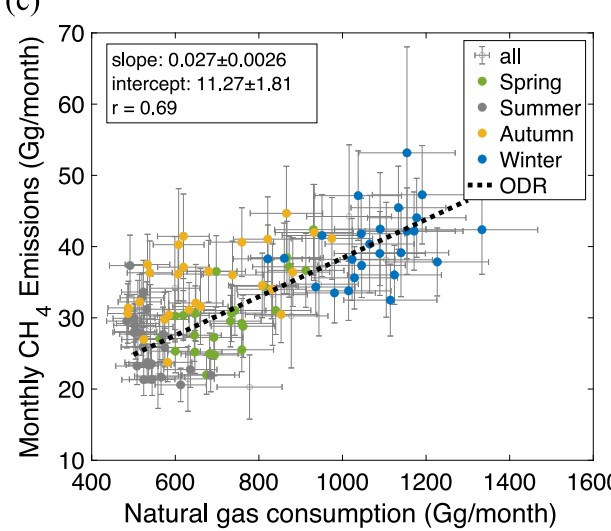

**Fig. 2 | Correlation between CH₄ emissions and natural gas consumption.**
**a** Time series of monthly $CH_4$ emissions (based on $CO_2$ emissions of ODIAC) from CLARS-FTS estimates (blue; left axis) and monthly natural gas consumption in the LA basin from residential, commercial and industrial sectors (red; right axis). The natural gas consumption time series has been shifted to the left by a half-month[19]. **b** The correlation between $CH_4$ emissions and natural gas consumption. The correlation coefficient is 0.69. Points are color-coded by season illustrating the progressive increase in emissions from summer (red) to winter (blue). A linear regression based on orthogonal distance regression (ODR), which considers the data uncertainty, is applied. The estimated slope and intercept are (2.9 ± 0.26)% and 10.56 ± 1.80 Gg/month, respectively. **c** The same as (**b**) for monthly $CH_4$ emissions estimated using CARB $CO_2$ inventory. The estimated slope and intercept are (2.7 ± 0.26) % and 11.27 ± 1.81 Gg/month, respectively. The error bars represent the estimation error (1σ) of the monthly values.

With regard to the uncertainty in the trend of the estimated $CH_4$ emissions, we see no significant trend in the excess ratio (Supplementary Fig. S8), indicating that $CH_4$ and $CO_2$ emissions are both declining at similar rates. Therefore, we can infer the $CH_4$ emission trend results from the $CO_2$ emission trend from bottom-up inventories. Although the absolute uncertainty of $CO_2$ bottom-up emissions is about 10% (ref. [28]), our knowledge of interannual variabilities in emissions and the corresponding decadal trend is better constrained because consistent methods are used to calculate bottom-up inventories for different years. Moreover, the good agreement of the $CO_2$ emissions trends derived from the CARB and ODIAC inventories suggests that the trend is reasonable, although their absolute values may vary.

## Comparison of top-down and bottom-up emissions trends
Under the terms of California legislation (Senate Bill 1371 enacted in 2014), gas companies are required to take feasible and cost-effective measures to avoid, reduce and repair natural gas leaks from their pipeline infrastructures. The mean annual decrease in emissions estimated by the gas utility is −5.8%/yr over 2015-2021 (see Methods), while the corresponding mean top-down trend estimated from the CLARS-FTS data is much smaller at approximately (−1.57 ± 0.41)%/yr. This discrepancy would be resolved if the 2015 baseline emissions value estimated by the gas utility was increased by a factor of 4 to about 138.2 Gg (i.e., 7.2 million Mscf; see Methods).

The assumption of this paper is that fugitive emissions come from leaks in the distribution system, including the distribution pipelines

### (a) CH₄ emissions based on the ODIAC CO₂ inventory

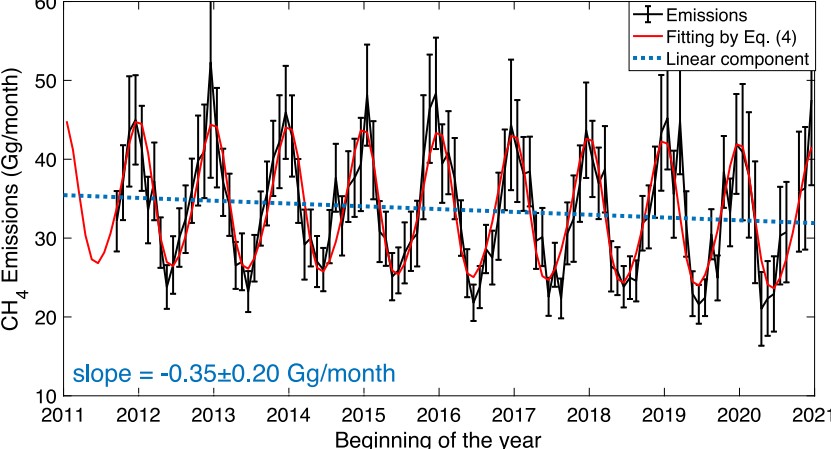

### (b) CH₄ emissions based on the CARB CO₂ inventory

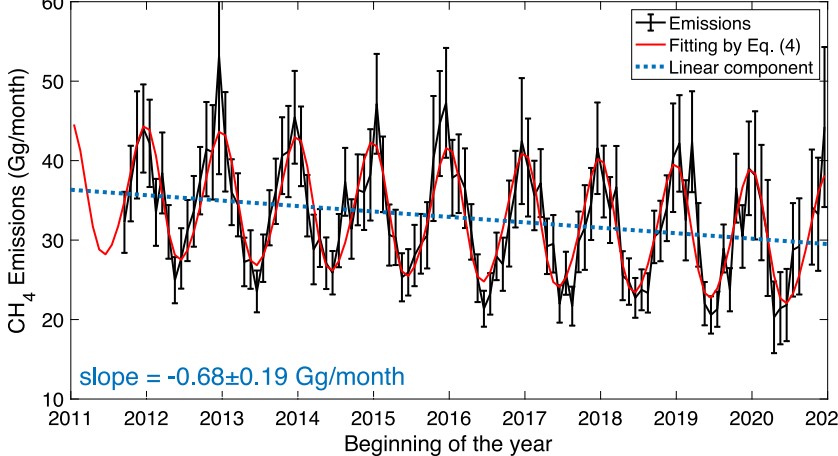

### (c) EEMD results

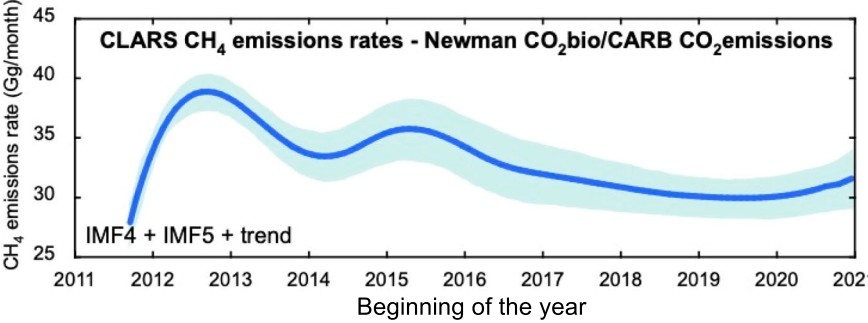

**Fig. 3 | Decadal decreasing trend of CH₄ emissions. a** Monthly CH₄ emissions, estimated based on the ODIAC CO₂ inventory, from Sept. 2011 to Dec. 2021 in the LA basin and fitting using a statistical model (Eq. (4)) that consists of a linear component and a seasonal component by harmonic functions. The linear component is extracted and shown in blue. The slope of this linear component is $-0.35 \pm 0.20$ Gg/month. **b** The same as (**a**) but for monthly CH₄ emissions, estimated based on the CARB CO₂ inventory. The slope of this linear component is $-0.68 \pm 0.19$ Gg/month. The error bars represent the estimation error (1σ) of the monthly emissions. **c** Interannual trend extracted from the Ensemble Empirical Mode Decomposition (EEMD) analysis based on the monthly CH₄ emissions estimated using the ODIAC CO₂ inventory. The beginning and end of the EEMD curves are influenced by edge effects for approximately a year at each end. The uncertainty band is ± 1σ.

and emissions in buildings beyond the meters. Current literature suggests that leaks from home and commercial buildings are not large enough to dominate the emissions indicated by the atmospheric measurements[29–32]. Fugitive emissions from residential appliances, furnaces plus water heaters plus stoves, contribute 13.4 Gg/yr (ref. [29,30,32,33]). If we use a conservative, high-end, estimate for fugitive commercial, industrial, and power plant emissions, combined, with double the residential emissions, then the total post-meter emissions in the LA Basin are 40.2 Gg/yr, only approximately 10% of the emissions that the CLARS measurements give. Therefore, the discrepancy must include emissions from a four-fold underestimation of the utility's baseline, raising it to about 138.2 Gg/yr, or overestimation of the utility's reported decrease in emissions, or a combination of these. Since overall CH₄ emissions in the LA basin have a significant contribution from natural gas fugitive emissions, this difference may have a significant impact on the attainability of the 40% reduction in statewide CH₄ emissions by 2030 mandated by California Senate Bill 1383 (ref. [19]). This depends on many other factors that drive CH₄ abatement including reduction of organic waste disposal, capture of methane from cattle manure and other efforts, but eliminating

fugitive emissions, in the extreme, would remove 5 of the total 39 million tonnes CH$_4$ as CO$_2$eq, based on the CARB inventory[34].

## Implications for emissions control policies

The decadal trend of CH$_4$ emissions in the LA basin from 2011 to 2020 is analyzed using remote sensing data from CLARS-FTS on Mt. Wilson, California. Over the decade, we estimated that about 50% of the variations in monthly CH$_4$ emissions can be explained by the natural gas usage from residential and commercial sectors. A fugitive emission rate of $(2.8 \pm 0.18)\%$ is obtained from the observed correlation between CH$_4$ emissions and natural gas consumption. The seasonal variabilities observed by CLARS-FTS are in good agreement with the long-term data from NOAA MWO flask measurements. The long-term CH$_4$ emissions showed a significant decreasing tend of $(-1.57 \pm 0.41)\%$/yr. Our results suggest that the current emissions control policies are effective. This study also highlights the importance of continuous observation and monitoring to verify the effectiveness of emissions reduction policies over the long term.

A key finding from this study is that the utility providing natural gas to the LA area may be significantly overestimating the magnitude of its methane emissions reductions, underestimating the extent of the baseline fugitive emissions from the natural gas infrastructure and end-users, or some combination of the two. This study shows that methane emissions control measures will likely need to be more aggressive to achieve the goal of reducing short-lived climate pollutants emissions by 40% by 2030 relative to the 2013 level in the Los Angeles basin. MRV strategies including the use of a multitiered observing system consisting of ground-based, aircraft, and satellite instruments[35] sensitive to point and area emissions sources, will be required to meet the objectives set forth in California legislation.

## Methods

### Converting XCO$_{2,xs}$ to XCO$_{2,ff}$ by correcting the biogenic fluxes in LA

A first-order correction is carried out for biogenic CO$_2$ fluxes to convert XCO$_{2,xs}$ to XCO$_{2,ff}$ by XCO$_{2,ff}$ = XCO$_{2,xs}$−XCO$_{2,bio}$, where XCO$_{2,xs}$ is the excess estimated from CLARS-FTS observations, and XCO$_{2,ff}$ and XCO$_{2,bio}$ are the contributions from fossil fuel and biogenic fluxes, respectively. We use the quantity XCO$_{2,ff}$ instead of XCO$_{2,xs}$ in the tracer-tracer inversion method to estimate CH$_4$ emissions. The monthly ratios (CO$_{2,ff}$/CO$_{2,xs}$) estimated from two sets of isotope measurements by Newman et al.[25] and Miller et al.[26], respectively, are used in this study, for comparison of the annual patterns (Supplementary Fig. S7). Since there is no significant interannual trend in the biogenic fluxes in LA (see Supplementary Text 3), we apply the monthly averaged ratios derived from all available data in the extended Newman et al.[25] data set (Supplementary Fig. S7). The time series of the excess ratio of XCH$_{4,xs}$/XCO$_{2,ff}$ after correcting the biogenic fluxes is shown in Supplementary Fig. S8.

### Estimating CH$_4$ emissions using CLARS-FTS observations

CLARS-FTS on Mt. Wilson, California, uses a pointing system to target a set of 33 predefined surface reflection points in the LA basin as well as a local diffuse reflector (Spectralon) for measurements of the free tropospheric background. CLARS-FTS surveys the whole basin every 1.5 to 2 h. Depending on the season and length of day, the entire basin is surveyed five to eight times per day. Column averaged dry-air mixing ratios of CH$_4$ (XCH$_4$), CO$_2$ (XCO$_2$), and other gases are retrieved from the reflected sunlight from the surface and the Spectralon[36]. This study applies the tracer-tracer inversion method[9,15] to estimate the monthly CH$_4$ emissions using CLARS-FTS observations from 2011 to 2020. First, excess XCH$_4$ (XCH$_{4,xs}$) and excess XCO$_2$ (XCO$_{2,xs}$) are calculated by subtracting the background values (XCH$_{4,BK}$ and XCO$_{2,BK}$), described

below, from the LA basin values (XCH$_{4,LA}$ and XCO$_{2,LA}$), respectively:

$$XCO_{2,xs} = XCO_{2,LA} - XCO_{2,BK} \tag{1}$$

$$XCH_{4,xs} = XCH_{4,LA} - XCH_{4,BK} \tag{2}$$

The background values (XCH$_{4,BK}$ and XCO$_{2,BK}$) are constructed by integrating the Spectralon retrievals (representing the backgrounds above the CLARS-FTS) and the NOAA MWO nighttime flask measurements[15] (representing the boundary layer backgrounds in LA; Supplementary Text 2). The XCO$_{2,xs}$ is then converted to XCO$_{2,ff}$ using the biogenic fluxes correction method described in Supplementary Text 3. Monthly CH$_4$ emissions (E$_{CH4}$) are then derived using the estimated monthly XCH$_{4,xs}$/XCO$_{2,ff}$ ratio:

$$E_{CH4}\big|_{monthly}^{top-down} = \frac{XCH_{4,xs}}{XCO_{2,ff}}\bigg|_{monthly}^{CLARS} \times E_{CO2}\big|_{monthly}^{inventory} \times \frac{MW_{CH4}}{MW_{CO2}} \tag{3}$$

where $E_{CO2}$ is CO$_2$ emissions (from CARB or ODIAC, discussed below), and $\frac{MW_{CH4}}{MW_{CO2}}$ is the ratio of the molecular weights of CH$_4$ (i.e., 16) and CO$_2$ (i.e., 44). This tracer-tracer inversion method is built on the strong correlations between XCH$_{4,xs}$ and XCO$_{2,xs}$ measured in the PBL in source regions. This method works because the lifetimes of both gases are much longer than the mixing time within the basin, and therefore the excess mixing ratios of both gases are highly correlated[9] (Supplementary Fig. S3), even though their sources are geographically distinct.

### CO$_2$ bottom-up inventory in the LA basin

Bottom-up CO$_2$ emissions are required to compute CH$_4$ emissions in Eq. (3). Several CO$_2$ emission estimates are available for the LA basin. Although Hestia is believed to have high accuracy (-10% for regional estimates[28]), the data are only available from 2010 to 2015. A longer estimate comes from ODIAC[37] at a 1-km spatial resolution that is available from 2000 to 2019. State-wide emissions in California[34] are publicly available from 2000 to 2020, but we only use through 2019, because of the effects in 2020 of the COVID-19 pandemic. In this study, we scaled the annual averages of ODIAC emissions by adding 3.5 TgCO$_2$/Month to match Hestia, as shown in Supplementary Fig. S2. The seasonal cycles of the scaled ODIAC inventory match Hestia very well. CARB monthly data are also produced by attributing the annual sum to all months based on monthly fractions from Hestia. For CO$_2$ emissions in 2020, we used the 2019 value as the baseline and applied scale factors derived from in-situ observations[38] to calculate the drawdown of CO$_2$ emissions in LA due to the COVID-19 pandemic lockdown. The 2020 reductions were $17\% \pm 9\%$, $34\% \pm 6\%$, and $28\% \pm 4\%$ in March, April, and May, respectively, relative to the 2019 levels[38]. For June, a 14% reduction (half of the reduction in May) is assumed. For the remaining months in 2020, a 5% reduction is assumed for each month based on our analysis of the reduction of traffic volumes in LA from the Caltrans Performance Measurement System (PeMS)[39]. The inventory uncertainty for every month after March 2020 is derived using error propagation from the baseline uncertainty (assumed to be 20%) and the estimation uncertainty[38]. This extrapolation is separately applied to ODIAC and CARB inventories, which are used to derive two sets of CH$_4$ emissions based on Eq. (3).

### Estimating the decreasing trend using linear regression

A statistical model that consists of a linear component and a seasonal component consisting of harmonic functions is fitted to the monthly CH$_4$ emissions from 2011 to 2020. The model is given by:

$$Emissions = \alpha_0 + \alpha_1 * t + \beta_1 * \sin(2\pi t) + \beta_2 * \cos(2\pi t) + \beta_3 * \sin(4\pi t) + \beta_4 * \cos(4\pi t) \tag{4}$$

where $\alpha_{0-1}$ are the coefficients for the linear component, and $\beta_{1-4}$ are the coefficients for the seasonal cycle component. The uncertainties for the slope in both cases (using ODIAC and CARB inventories) are estimated using the Monte Carlo method, which samples the monthly emissions using a normal distribution based on the mean and error and estimates the slope. The method makes 10,000 simulations for the emission time series and obtains the standard deviation of the slope samplings. The uncertainties for the slope in both cases are estimated using the Monte Carlo method, which samples the monthly emissions using a normal distribution based on the mean and error and estimates of the slope. The histograms of sample slopes are shown in Supplementary Fig. 12.

### Ensemble empirical mode decomposition (EEMD) analysis

EEMD is a powerful tool for extracting trend information from non-linear and nonstationary time series[25,40,41]. The method breaks down the time series into intrinsic mode functions (IMFs). The IMFs have increasing period lengths and the final one is a long-term trend with at most only one minimum or maximum. High-frequency modes are generated first, with the earliest mode representing noise. The later modes (e.g., IMFs 3 and 4) are interpreted in terms of known processes such as annual cycles. An ensemble of 300 time series is generated by adding random noise equivalent to the error in the measurements, following Wu and Huang[42]. EEMD analyses are applied to the ensemble time series and the outputs are averaged. The EEMD technique is a data adaptive technique without assumptions on the shapes of the IMFs. The results are shown in Supplementary Figures S14, S15 for $CH_4$ emissions (corrected using biospheric fluxes in Newman et al.[25]) using ODIAC and CARB $CO_2$ emissions, respectively. The beginning and end of the EEMD curves are influenced by edge effects, for approximately a year at each end.

### $CH_4$ emissions by the gas utility

The California legislation Senate Bill 1371 (enacted in 2014) requires natural gas utilities to avoid, reduce and repair gas leaks emanating from their pipeline infrastructures. This measure additionally requires the development of compliance plans, updated every two years, that report their best estimates of system-wide emissions and projected reductions using 2015 as a baseline year. The gas utility that serves the greater LA area estimated its baseline emissions by accounting for estimated and modeled emissions from system components including pipelines, compressor stations, customer meters, underground storage, and other leak sources. Their most recent annual report in 2022 (ref. [43,44]) estimated their system's 2015 baseline emissions to be 1,797,141 Mscf (1 Mscf ≡ 1000 scf; The M refers to the Roman numeral for thousand), which is about 34.5 Gg/year. However, this is roughly 1/10 of our estimates of the $CH_4$ emissions in LA. The report further estimated the system's total annual volume of leaks and emissions to be 1,309,873 Mscf (25.1 Gg) and 1,129,467 Mscf (21.7 Gg) in 2020 and 2021, respectively. The average annual decreases in emissions estimated by the gas utility are therefore 5.4%/yr and 6.2%/yr, respectively, by comparing the emissions in 2020 and 2021 to the baseline in 2015. The average rate of decrease is −5.8%/yr. The uncertainties in these estimations are not provided but one indication is the variability in the 2015 baseline values emissions values which range from a low of 1,797,141 Mscf (34.5 Gg) reported in the revised 2022 annual report to a high of 2,779,853 Mscf (53.4 Gg) reported in the revised 2018 report[45], a difference of 35%. This result suggests that the decrease rate reported by the gas company is larger than CLARS-FTS observations, and the absolute leakage mass may have been significantly underestimated.

### Data availability

CLARS-FTS $XCO_2$ and $XCH_4$ data are publicly available at https://data.caltech.edu/records/254mc-zpg74 (https://doi.org/10.22002/D1.1985). NOAA carbon cycle surface flask measurements on Mt. Wilson are available from https://gml.noaa.gov/dv/site/site.php?code=MWO and can be requested from NOAA Earth System Research Laboratories. Bottom up inventory of $CO_2$ emissions from ODIAC are publicly available from https://db.cger.nies.go.jp/dataset/ODIAC/, from California Air Resources Board are available from https://ww2.arb.ca.gov/ghg-inventory-data, and from Hestia v2.5 are publicly available from https://hestia.rc.nau.edu/; The reports of quarterly natural gas usage are publicly available from SoCalGas at https://www.socalgas.com/for-your-business/energy-savings/energy-usage-requests.

### Code availability

The EEMD codes (in Matlab) used to determine the $CH_4$ emissions trends are located in the CaltechData repository at https://doi.org/10.22002/5f3rd-xqr42.

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

## Acknowledgements
Flask data provided by Dr. Ed Dlugokencky and Dr. Arlyn Andrews (NOAA) are greatly appreciated. Comments from K.-F. Li and J. Pinto are gratefully acknowledged. Funding is acknowledged from NASA grant 80NSSC21K1929 (T.O.) and NIST grant 70NANB19H129 (K. G.) P.I.P. acknowledges support from the UK National Centre for Earth Observation (NCEO) funded by the Natural Environment Research Council (NE/R016518/1). S.S. and T.P. acknowledge support for the CLARS facility from the NASA Earth Science Directorate and the JPL Earth Science and Technology Directorate. The research was carried out, in part, at the Jet Propulsion Laboratory, California Institute of Technology, under a contract with the National Aeronautics and Space Administration (80NM0018D0004)

## Author contributions
S.S. and Z.-C.Z designed the study. Z.-C. Z., T.P., S.S. and S.N. carried out the experiments. S.S., Z.-C.Z., P.P., Y.L.Y. and S.N. analyzed the results. T.O. and K.G. provided carbon dioxide emission inventories. Z.-C.Z. and S.S. wrote the paper. All authors reviewed the manuscript.

## Competing interests
The authors declare no competing interests.
