## [Peer Review File · Nature Communications]

Decadal decrease in Los Angeles methane emissions is much smaller than bottom-up estimatesReviewer #1 (Remarks to the Author):

Review of "Decrease in Los Angeles methane emissions is much smaller than bottom-up estimates," by Z.-C. Cheng and colleagues

This is an interesting follow-on study to previous ones from this group using a mountain top spectrometer to estimate methane emissions from the Los Angeles area. The manuscript estimates regional emissions over nearly a decade (2011-2020), and finds a decrease of roughly 1.5% per year over the time period. The group has previously published shorter parts of this dataset using the same emission calculation technique (e.g., using the measured ratio between methane and CO₂ to estimate emissions using a CO₂ inventory) and relating seasonality in methane to natural gas emissions, but here they show a longer record. This is a valuable, unique long-term dataset which importantly shows a downward trend in methane emissions across a large city in a domain with strong climate mitigation policy. At the same time, though, it only provides an incremental advance beyond what the group has previously published, with the only discernable difference being the length of the dataset. I hoped for more detail to convincingly link the findings to policy changes or bottom up emissions, but did not find it in this version of the manuscript.

I also have some questions about the methods that were not addressed in the brief text. I was surprised that the text did not mention the possible impact of biogenic CO₂ fluxes on estimated methane emissions, given that the technique used here relies on observations of atmospheric CO₂ that are affected by fossil and biogenic fluxes. Recent work in Los Angeles (Miller et al. 2020, <https://doi.org/10.1073/pnas.2005253117>) demonstrated that the biosphere leaves a measurable imprint on CO₂ excess levels equivalent to 1/3 of fossil fuel emissions, with net uptake of CO₂ in summer by the biosphere. Isn't it possible that biogenic CO₂ fluxes will affect the inferred seasonal cycle of methane? Even if biogenic fluxes do not affect the emissions or seasonality of methane, it is odd that this is not addressed in the text.

The other main issue with the papers is the lack of detailed, quantitative accounting of methane emissions from natural gas and other possible sources. The manuscript asserts that the declining trend is due to climate mitigation policies (lines 127-130), however does not discuss any actual changes to sources in the region apart from what is reported by the gas utility. I found the presented analysis and discussion of the gas utility data on lines 342-356 to be insufficiently quantitative—it is difficult to compare reductions in natural gas emissions on a percent basis when its only part of the total emissions in the basin. The authors should attempt to convert these estimates into emission units that make the changes to gas emissions comparable to overall emission estimates from their remote sensing technique. In addition, no other methane emission sources apart from natural gas leaks are discussed here, despite previously published studies that detect reductions in Los Angeles methane emissions due to changes in landfill emissions (Yadav et al. 2019 <https://doi.org/10.1029/2018JD030062> and Yadav et al. 2023 DOI:10.1088/1748-9326/acb6a9). This seems like something the authors could potentially address by looking in more detail at the trends at different receptor locations in the basin, but was not mentioned in the study.

In sum, this is an interesting and valuable dataset but the analysis lacked support for the bold claims made in the title and text.

Detailed comments:

Abstract:

The abstract states, "Targets set at all levels of government call for reducing anthropogenic methane emissions by at least 30 percent by 2030 from 2020 levels." (lines 23-25). This is in fact incorrect. CARB's 2022 Scoping Plan says, "All SB 1383 emissions reductions are mandated to be realized by 2030 and are relative to 2013 levels."

Lines 95-97: "One exception is the excess ratio in March of 2019, which is found to be related to lower temperature and higher demand in the natural gas supply." Where is the data that back this assertion up? Please provide a citation or supplementary figure with this analysis. Spring of 2020 was anomalous because of reduced CO₂ emissions, making the ratio increase, and yet calculated CH₄ emissions appear anomalously low. How certain are you that CH₄ emissions actually did decline? Or is that just due to the uncertainty in CO₂ decreases?

Lines 100-101: "However, we noted that the estimated CH₄ emissions in Boston also showed a decrease in April, 2020." What might be the cause, and why might we expect Boston to follow similar trends as LA?

Lines 101-102: "decreasing interannual trend of the emissions that occurs in the second half of the year which drives the decreasing trend..."
Is there a statistical test done that supports this assertion?

Lines 131-136: "The goal of reducing emissions of short-lived climate pollutants by 2030 relative to the 2013 level can be achieved by capturing or avoiding methane emissions from a variety of sources including dairy manure, enteric fermentation, disposal of organics at landfills, and fugitive methane emissions²⁹. The observed decrease in CH₄ emissions inferred from CLARS-FTS measurement demonstrates the effectiveness of California legislation beginning with AB 32, the Global Warming Solutions Act, in 2006." What changes in methane emissions did the legislation bring about? Do we have quantitative estimates of what these might be? Have you compared these trends to what the California methane inventory shows?

Lines 156-7: "The average annual decreases in emissions estimated by the gas utility are 5.4 %/yr and 6.2 %/yr, respectively." What are these two different percents referring to? After re-reading this several times, I guess these are the two different CO₂ inventories used. However, it is very unclear as currently written.

Lines 157-9: "The average top-down trend estimated from the CLARS data, approximately -1.5 %/yr is considerably smaller than the approximately -6 %/yr inferred from the bottom-up estimates from the gas utility." ♦ It is difficult to understand this argument presented as percentages. Would be easier to understand as emissions numbers.

Lines 179-180: "A key finding from this study is that the utility providing natural gas to the Los Angeles area may be significantly overestimating the magnitude of its methane emissions reductions," I don't think there is enough detail or analysis presented here to make this claim. Authors should add more detail to the analysis of natural gas data to support this assertion.

Reviewer #2 (Remarks to the Author):

This manuscript describes a decade of methane emission estimates from the LA Basin based on the ratio of excess methane and carbon dioxide concentrations. The seasonal cycle in emissions and correlation with natural gas consumption is analyzed to determine the natural gas leak rate. The authors find a trend in decreasing methane emissions, but substantially less than reported by the natural gas utility. I think this is a valuable manuscript that would be of interest to Nature Communications readers, but minor edits are needed to improve clarity. I suggest adding more information discussing the following issues:

1. What is the cause of the seasonal trend in natural gas consumption? Residential and commercial heating with natural gas in the winter? Why would fugitive emissions increase with increased consumption? Higher pipeline pressure?
2. Clarify what is the non-seasonal component of methane emissions? Does this include non-natural gas sources like landfills? What about oil and gas production? How does this estimate compare to the CARB inventory?
3. Directly compare the absolute emissions and loss rate that you calculate to the utility's estimate for the start and end of the time series. Mention some of the mitigation actions taken by SoCalGas to reduce methane emissions.

Item-by-item responses to the specific comments are provided below, in which the reviewers' comments are in **blue**, our responses in **black**, and modifications of the original manuscript are indicated by highlight in **yellow**.

REVIEWER COMMENTS

Reviewer #1 (Remarks to the Author):

Review of “Decrease in Los Angeles methane emissions is much smaller than bottom-up estimates,” by Z.-C. Cheng and colleagues

This is an interesting follow-on study to previous ones from this group using a mountain top spectrometer to estimate methane emissions from the Los Angeles area. The manuscript estimates regional emissions over nearly a decade (2011-2020), and finds a decrease of roughly 1.5% per year over the time period. The group has previously published shorter parts of this dataset using the same emission calculation technique (e.g., using the measured ratio between methane and CO₂ to estimate emissions using a CO₂ inventory) and relating seasonality in methane to natural gas emissions, but here they show a longer record. This is a valuable, unique long-term dataset which importantly shows a downward trend in methane emissions across a large city in a domain with strong climate mitigation policy. At the same time, though, it only provides an incremental advance beyond what the group has previously published, with the only discernable difference being the length of the dataset. I hoped for more detail to convincingly link the findings to policy changes or bottom up emissions, but did not find it in this version of the manuscript.

An ensemble empirical mode decomposition (EEMD) analysis was carried out to determine the interannual trend from the derived methane emissions time series and we related the trend to changes in policies. EEMD is a powerful tool for extracting trend information from nonlinear and nonstationary time series. An introduction to EEMD has been added to the **Methods** section in the revised paper.

The interannual trend of CH₄ emissions extracted from the EEMD analysis shows a larger drop starting around 2015. This inflection point in emissions occurs around the years when the provisions of California Senate Bill (SB)1371 (approved on September 21, 2014) and SB1383 (approved on September 19, 2016) came into effect. SB1383 requires the state Air Resources Board to develop and begin implementing comprehensive strategy to reduce emissions of short-lived climate pollutants, for methane to achieve a reduction by 40% below 2013 levels by 2030. SB1371 specifically targets reducing natural gas leakage from the Public Utilities Commission-regulated gas pipeline facilities that are intrastate transmission and distribution lines. With the approval of these bills, it is reasonable that the most rapid progress would have been made in the first few years after 2015 because it's easier to find and stop large methane leaks compared to smaller leaks.

Related reference:

Huang, N., Shen, Z., and Long, S.: The empirical mode decomposition and the Hilbert spectrum for nonlinear and non-stationary time series analysis, *Proc. R. Soc. Lon, Ser.-A*, 454, 903–995, 1998.

Kobayashi-Kirschvink, K. J., Li, K.-F., Shia, R.-L., and Yung, Y. L.: Fundamental modes of atmospheric CFC-11 from empirical mode decomposition, *Adv. Adapt. Data Anal.*, 4, 1250024, doi:10.1142/S1793536912500240, 2012.

Wu, Z. and Huang, N. E.: Ensemble empirical mode decomposition: A noise-assisted data analysis method, *Adv. Adapt. Data Anal.*, 1, 1–41, 2009.

California Legislature, 2016, SB-1383 Short-lived climate pollutants: methane emissions, dairy and livestock, organic waste, landfills, https://leginfo.legislature.ca.gov/faces/billNavClient.xhtml?bill_id=201520160SB1383 accessed 28 November 2022.

California Legislature, 2014, SB-1371 Natural gas: leakage abatement, https://leginfo.legislature.ca.gov/faces/billTextClient.xhtml?bill_id=201320140SB1371 accessed 28 November 2022.

I also have some questions about the methods that were not addressed in the brief text. I was surprised that the text did not mention the possible impact of biogenic CO₂ fluxes on estimated methane emissions, given that the technique used here relies on observations of atmospheric CO₂

that are affected by fossil and biogenic fluxes. Recent work in Los Angeles (Miller et al. 2020, <https://doi.org/10.1073/pnas.2005253117>) demonstrated that the biosphere leaves a measurable imprint on CO₂ excess levels equivalent to 1/3 of fossil fuel emissions, with net uptake of CO₂ in summer by the biosphere. Isn't it possible that biogenic CO₂ fluxes will affect the inferred seasonal cycle of methane? Even if biogenic fluxes do not affect the emissions or seasonality of methane, it is odd that this is not addressed in the text.

In the revised manuscript, the impacts of biogenic CO₂ fluxes on the methane emission estimation are explicitly considered. We re-run the calculation of excess ratio using the XCO_{2,ff}, derived as the product of XCO_{2,xs} and CO_{2,ff}/CO_{2,xs}. The monthly ratios (CO_{2,ff}/CO_{2,xs}) are estimated from two sets of isotope measurements by Newman et al. (2016) and Miller et al. (2020), respectively.

(1) The impact on the seasonal cycle

In Newman et al. (2016), which used the ¹⁴CO₂ data to put better constraints on contributions from anthropogenic emissions and the biosphere to the observed CO₂ enhancement in the Los Angeles Basin, it was found that the maximum biosphere contribution was during winter 2012–2013, 7 ppm (28 % of the total C_{ff}), and the minimum was 0.1 ppm during spring of 2010. The average is (4.1 ± 0.5) ppm (16 % of C_{ff}) during cooler months and (2.2 ± 0.3) ppm (8 % of C_{ff}) during warmer months. An extension of the time series is shown in **Supplementary Figure S6**. In Miller et al. (2020), which used measurements of Δ14C and CO₂ to separate biogenic and fossil contributions to CO₂ enhancements above background, found that the urban biospheric component is a source in winter and a sink in summer, with an estimated amplitude of 4.3 parts per million (ppm), equivalent to 33% of the observed annual mean fossil fuel contribution of 13 ppm. The CO_{2,xs}/CO_{2,ff} -1.0 monthly mean data are shown in **Supplementary Figure S7**.

The higher values in winter than summer affect the seasonality of the methane emission estimations in this study. If the biogenic contributions are considered, the seasonality will be enhanced, leading to even larger seasonal amplitude. Because the summer time XCO_{2,xs} from anthropogenic emissions is underestimated, while that in the winter is overestimated, considering the biospheric uptake will lead to even smaller XCH_{4,xs}/XCO_{2,xs} in summer and larger excess ratio in winter.

The resulting time series of excess ratio of XCH_{4,xs}/XCO_{2,ff} are shown in **Supplementary Figure S8**. The correlation of NG usages and CH₄ emissions based on the updated excess ratio are shown in **Figure 3** and **Supplementary Figure S9**. We can see that, using biogenic fluxes in Newman et al. (2016), the slopes from the linear fit, an indicator of NG loss rates, have increased to 2.6% and 2.9% based on ODIAC and CARB inventories, respectively. For biogenic fluxes in Miller et al. (2020), the slopes are 3.7% and 3.5%, respectively. Both of these sets of estimates are at the lower bounds of the estimates by Wennberg et al. (2012), which showed a loss rate of approximately 2.5–6% of the natural gas delivered to basin customers. The decreasing trends analysis have also been added in **Supplementary Figure S10**.

(2) The impact on the inter-annual trend:

The following figure (**Supplementary Figure S6**), extended from the results in Newman et al. (2016), show biogenic contribution from 2006 and mid-2016 (data after 2017 are not available).

The slope of the CO_2bio trend is 0.013 ppm/year, which is very small and indicates no significant trend. Based on this result, it is assumed the biogenic fluxes in the basin do not have a significant interannual trend that could affect the derived methane emissions in the paper.

Supplementary Figure S6. The contribution from biosphere and fossil fuel constrained from ¹⁴CO₂ data, a time series extended from Newman et al. (2016). The slope of the CO₂bio trend is 0.013 ppm/year from linear regression analysis.

Supplementary Figure S7. Monthly mean of $(C_{xs}/C_{ff} - 1.0)$ inferred from ¹⁴CO₂ data by Newman et al. (2016) and Miller et al. (2020), respectively.

Supplementary Figure S8. Excess ratio of XCH_4xs/XCO_2xs before and after correcting the biogenic fluxes.

(a) Newman et al. 2016 + CARB

(b) Newman et al. 2016 + ODIAC

(c) Miller et al. 2020 + CARB

(d) Miller et al. 2020 + ODIAC

Supplementary Figure S9. Correlation of NG and CH_4 emissions using different data for biogenic flux correction (Newman et al. 2016 and Miller et al. 2020) and different CO_2 inventories (CARB and ODIAC).

(a) NewmanBio+ ODIAC

(b) NewmanBio + CARB

(c) MillerBio + ODIAC

(d) MillerBio + CARB

Supplementary Figure S10. Decreasing trends using the updated CH₄ emissions, the decreasing slopes are also indicated.

Reference added:

[1] Newman, S., Xu, X., Gurney, K. R., Hsu, Y. K., Li, K. F., Jiang, X., Keeling, R., Feng, S., O'Keefe, D., Patarasuk, R., Wong, K. W., Rao, P., Fischer, M. L., and Yung, Y. L.: Toward consistency between trends in bottom-up CO₂ emissions and top-down atmospheric measurements in the Los Angeles megacity, *Atmos. Chem. Phys.*, 16, 3843–3863, <https://doi.org/10.5194/acp-16-3843-2016>, 2016.

[2] Miller et al., Large and seasonally varying biospheric CO₂ fluxes in the Los Angeles megacity revealed by atmospheric radiocarbon, *PNAS*, <https://doi.org/10.1073/pnas.2005253117>

The other main issue with the papers is the lack of detailed, quantitative accounting of methane emissions from natural gas and other possible sources. The manuscript asserts that the declining trend is due to climate mitigation policies (lines 127-130), however does not discuss any actual changes to sources in the region apart from what is reported by the gas utility. I found the presented analysis and discussion of the gas utility data on lines 342-356 to be insufficiently quantitative—it is difficult to compare reductions in natural gas emissions on a percent basis when its only part of the total emissions in the basin. The authors should attempt to convert these estimates into emission units that make the changes to gas emissions comparable to overall emission estimates from their remote sensing technique. In addition, no other changes methane emission sources apart from natural gas leaks are discussed here, despite previously published studies that detect reductions in Los Angeles methane emissions due to changes in landfill emissions (Yadav et al. 2019 <https://doi.org/10.1029/2018JD030062> and Yadav et al. 2023 DOI:10.1088/1748-9326/acb6a9). This seems like something the authors could potentially address by looking in more detail at the trends at different receptor locations in the basin, but was not mentioned in the study.

(1) Emission unit conversion

For the SoCalGas leakage emissions, we do a conversion from the unit of SCF (standard cubic feet) to Gg (giga-gram) for methane emissions. For each SCF, there is 1.2 mole natural gas; for each mole natural gas, there is 16g CH₄ (if we assume all is methane in NG, approximately). For the 1797141 MSCF/year, that is about 34.5 Gg/year, which is about 1/10 of our estimates of leakage. This result suggests that the decrease rate reported by the gas company is larger than CLARS observations but the absolute leakage mass is significantly underestimated.

(2) Discussions on the changes of methane emission in the basin

In the LA basin, important methane sources include landfills, dairy farms, wastewater treatment, fossil fuel extraction and distribution, and fugitive emissions. As suggested by Yadav et al. (2023), landfills and natural gas infrastructure are the most likely plausible sources of emissions reductions in the LA basin. The Sunshine Canyon Landfill, the largest landfill in the LA basin, and its emissions have declined due to improved management practices (e.g., Cusworth et al 2020).

From **Supplementary Figure S13**, the timeseries of $XCH_{4,xs}/XCO_{2,xs}$ for three subregions in the basin, we can see there are no significant spatial differences. The air in the basin quickly mixed from the perspective of column abundances as measured by CLARS, which has a long light path in the basin extended from the surface target to the instrument. The CLARS observation geometry makes it sensitive to basin-wide changes in methane emissions, however, and less sensitive to any

specific target. Therefore, trends from individual receptor sites do not allow identification of particular stationary sources of emissions.

Supplementary Figure S13. The time series of $XCH_{4,xs}/XCO_{2,xs}$ ratio averaged for the three sub-regions, including western, central and eastern LA in the basin, to investigate the spatial patterns of the excess ratio.

In sum, this is an interesting and valuable dataset but the analysis lacked support for the bold claims made in the title and text.

Detailed comments:

Abstract:

The abstract states, “Targets set at all levels of government call for reducing anthropogenic methane emissions by at least 30 percent by 2030 from 2020 levels.” (lines 23-25). This is in fact

incorrect. CARB's 2022 Scoping Plan says, "All SB 1383 emissions reductions are mandated to be realized by 2030 and are relative to 2013 levels."

We have double checked the CARB's 2022 Scoping Plan and the SB 1383, and have revised the statements to "Targets set at all levels of government call for reducing anthropogenic methane emissions by 40 percent by 2030 from 2013 levels."

SB 1383.

https://leginfo.legislature.ca.gov/faces/billTextClient.xhtml?bill_id=201520160SB1383

The bill says:

This bill would require the state board, no later than January 1, 2018, to approve and begin implementing that comprehensive strategy to reduce emissions of short-lived climate pollutants to achieve a reduction in methane by 40%, hydrofluorocarbon gases by 40%, and anthropogenic black carbon by 50% below 2013 levels by 2030, as specified. The bill also would establish specified targets for reducing organic waste in landfills.

Lines 95-97: "One exception is the excess ratio in March of 2019, which is found to be related to lower temperature and higher demand in the natural gas supply." Where is the data that back this assertion up? Please provide a citation or supplementary figure with this analysis. Spring of 2020 was anomalous because of reduced CO₂ emissions, making the ratio increase, and yet calculated CH₄ emissions appear anomalously low. How certain are you that CH₄ emissions actually did decline? Or is that just due to the uncertainty in CO₂ decreases?

In the revised manuscript, we have removed this statement related to the excess ratio in March of 2019. Because of the uncertainty in the estimated excess ratio and emissions, it is not conclusive in this case to compare emissions and the driving factor (e.g., temperature) for a single month.

The excess ratio estimated by CLARS shows a clear increase compared to previous years. However, the estimated CH₄ emissions for Spring of 2020 is lower than previous years, primarily driven by the assumptions on the declined CO₂ emissions in 2020. This CO₂ emission drawdown was estimated by ground-based observations in Los Angeles. Therefore, judging from the difference and the uncertainty by the error bar, the decrease in CH₄ in April of 2020 is lower than previous years by more than one standard deviation (the error bar).

This decrease in CH₄ in April of 2020 is consistent with the decrease in Boston in the same month. Please see our responses to the next comment.

Lines 100-101: "However, we noted that the estimated CH₄ emissions in Boston also showed a decrease in April, 2020." What might be the cause, and why might we expect Boston to follow similar trends as LA?

Evidence has shown that Boston and Los Angeles may have large sources of methane from end user appliances fugitive emissions.

As suggested by **Sargent et al. (2021)**, the marked decrease in methane emissions at BU could be due to reduced appliance use in office buildings, restaurants, and/or the BU campus surrounding the BU site. The significant change in methane emissions at BU during the Covid-19 shutdown indicates that changes in local consumption are driving methane enhancements. The significant decrease in CH₄ emissions locally around BU during April 2020, when residential NG consumption and pipeline losses were constant, points to the importance of other sources such as beyond-the-meter losses and the necessity of further studies to quantify these sources.

Reference:

Sargent, M.R., Floerchinger, C., McKain, K., Budney, J., Gottlieb, E.W., Hutyra, L.R., Rudek, J. and Wofsy, S.C., 2021. Majority of US urban natural gas emissions unaccounted for in inventories. Proceedings of the National Academy of Sciences, 118(44).

Lines 101-102: “decreasing interannual trend of the emissions that occurs in the second half of the year which drives the decreasing trend...” Is there a statistical test done that supports this assertion?

From **Supplementary Figure S11**, we can see that the emissions from July to December has dominating the contribution to the interannual decline in methane emissions. The cause is still not clear and needs further investigation.

Lines 131-136: “The goal of reducing emissions of short-lived climate pollutants by 2030 relative to the 2013 level can be achieved by capturing or avoiding methane emissions from a variety of sources including dairy manure, enteric fermentation, disposal of organics at landfills, and fugitive methane emissions. The observed decrease in CH₄ emissions inferred from CLARS-FTS measurement demonstrates the effectiveness of California legislation beginning with AB 32, the

Global Warming Solutions Act, in 2006.” What changes in methane emissions did the legislation bring about? Do we have quantitative estimates of what these might be? Have you compared these trends to what the California methane inventory shows?

The mitigation actions taken by the government, including SoCalGas, has been summarized in SB 1371 and SB 1383, which are two bills by the California government to cut methane emissions. The bill SB1383 requires the state board to approve and begin implementing comprehensive strategy to reduce emissions of short-lived climate pollutants, and for methane to achieve a reduction by 40% below 2013 levels by 2030. The measures include adopting regulations to reduce methane emissions from livestock manure management operations and dairy manure management operations and to reduce the landfill disposal of organics. The bill 1371 specifically targets reducing natural gas leakage from the commission-regulated gas pipeline facilities that are intrastate transmission and distribution lines.

The CARB inventory for CH₄ is only available for the whole California state, and not available for Los Angeles. Unfortunately, a derivation of the CH₄ emissions in Los Angeles based on the California emission is not trivial. The method of scaling by population that works well for CO₂ is not applicable to CH₄, since CH₄ has much more complex sources in California that are not proportional to population.

Methane emission inventories in California by CARB:

https://ww2.arb.ca.gov/sites/default/files/classic/cc/inventory/data/tables/ghg_inventory_sector_sum_2000-19ch4.pdf

Information about SB1371 and SB1383:

[1] SB 1371 (Natural gas: leakage abatement) was approved by Governor on September 21, 2014: https://leginfo.legislature.ca.gov/faces/billTextClient.xhtml?bill_id=201320140SB1371

[2] SB 1383 (Short-lived climate pollutants: methane emissions: dairy and livestock: organic waste: landfills) was approved by Governor on September 19, 2016:

https://leginfo.legislature.ca.gov/faces/billTextClient.xhtml?bill_id=201520160SB1383

Lines 156-7: “The average annual decreases in emissions estimated by the gas utility are 5.4 %/yr and 6.2 %/yr, respectively.” What are these two different percents referring to? After re-reading this several times, I guess these are the two different CO₂ inventories used. However, it is very unclear as currently written.

The two numbers are derived from comparing the emissions in 2020 (1,309,873 Mscf) and 2021 (1,129,467 Mscf) relative to the baseline in 2015 (1,797,141 Mscf). We have rephrased the statements to be:

The average annual decreases in emissions estimated by the gas utility are therefore 5.4 %/yr and 6.2 %/yr, respectively, by comparing the emissions in 2020 and 2021 relative to the baseline in 2015.

Lines 157-9: “The average top-down trend estimated from the CLARS data, approximately -1.5

%/yr is considerably smaller than the approximately -6 %/yr inferred from the bottom-up estimates from the gas utility.” à It is difficult to understand this argument presented as percentages. Would be easier to understand as emissions numbers.

For the SoCalGas leakage emissions, we do a conversion from the unit of scf (standard cubic feet) to Gg (giga-gram) for methane emissions. For the 1797141 MSCF/year, that is about 34.5 Gg/year, which is about 1/10 of our estimates of leakage. This result shows that the decrease rate reported by the gas company is larger than CLARS observations but the absolute leakage mass is significantly underestimated.

Lines 179-180: “A key finding from this study is that the utility providing natural gas to the Los Angeles area may be significantly overestimating the magnitude of its methane emissions reductions,” I don’t think there is enough detail or analysis presented here to make this claim. Authors should add more detail to the analysis of natural gas data to support this assertion.

More details related to the analysis of CH₄ emissions and the comparison with natural gas have been added, including:

- (1) We analyze the trend using EEMD, and related the decreasing trend to the implementation of SB1383 and SB1371. We found that the interannual trend of CH₄ emissions extracted from the EEMD analysis shows a larger drop starting around 2015. This inflection point in emissions occurs around the years when the provisions of SB1371 (approved on September 21, 2014) and SB1383 (approved on September 19, 2016) came into effect.
- (2) We changed the unit to emissions, and compare the emissions with our estimates. This result shows that the decrease rate reported by the gas company is larger than CLARS observations but the absolute leakage mass is significantly underestimated.

Reviewer #2 (Remarks to the Author):

This manuscript describes a decade of methane emission estimates from the LA Basin based on the ratio of excess methane and carbon dioxide concentrations. The seasonal cycle in emissions and correlation with natural gas consumption is analyzed to determine the natural gas leak rate. The authors find a trend in decreasing methane emissions, but substantially less than reported by the natural gas utility. I think this is a valuable manuscript that would be of interest to Nature Communications readers, but minor edits are needed to improve clarity. I suggest adding more information discussing the following issues:

1. What is the cause of the seasonal trend in natural gas consumption? Residential and commercial heating with natural gas in the winter? Why would fugitive emissions increase with increased consumption? Higher pipeline pressure?

The seasonal trend in natural gas consumption is caused by the high demand by the residential sector (He et al., 2019), primarily for space heating. The correlation between natural gas consumption and CH₄ emissions may be due to increased wintertime demand by appliances for space heating, water heating, cooking, and other purposes that involve heat generation. The fugitive emissions may come from gas-fired appliances and industrial combustors under different operating conditions (start-up, operation, and shut-down). The pipeline pressure may be kept constant along the year, but the fugitive emissions may be increased from end-user appliances with increasing demand for heating in winter. There is also increasing evidence that the probability density functions for CH₄ emissions have a long tail, characterized by a small number of emitters with very large emissions, perhaps due to malfunctioning equipment or improper operating conditions (Zavala-Araiza et al., 2015).

References:

- [1] Zavala-Araiza, D., Lyon, D., Alvarez, R. A., Palacios, V., Harriss, R., Lan, X., et al. (2015). Toward a functional definition of methane super-emitters: Application to natural gas production sites. *Environmental Science & Technology*, 49(13), 8167–8174. <https://doi.org/10.1021/acs.est.5b00133>
- [2] He, L., Zeng, Z.-C., Pongetti, T., Wong, C., Liang, J., Gurney, K. R., et al. (2019). Atmospheric methane emissions correlate with natural gas consumption from residential and commercial sectors in Los Angeles. *Geophysical Research Letters*, 46. <https://doi.org/10.1029/2019GL083400>

2. Clarify what is the non-seasonal component of methane emissions? Does this include non-natural gas sources like landfills? What about oil and gas production? How does this estimate compare to the CARB inventory?

The non-seasonal component (the intercept) gives the CH₄ emissions extrapolated to zero metered consumption, that is, associated with non-metered emissions. The latter would include emissions from landfills, waste-water treatment, local geological sources, and natural gas transmission lines and mains. These sources may have their own seasonality that this simple two-parameter model cannot capture.

The CARB inventory for CH₄ is only available for the whole California state, and not available for Los Angeles. Unfortunately, a derivation of the CH₄ emissions in Los Angeles based on the California emission is not trivial. The method of scaling by population that works well for CO₂

is not applicable to CH₄, since CH₄ has much more complex sources in California that are not proportional to population.

3. Directly compare the absolute emissions and loss rate that you calculate to the utility's estimate for the start and end of the time series. Mention some of the mitigation actions taken by SoCalGas to reduce methane emissions.

For the SoCalGas leakage emissions, we do a conversion from the unit of scf (standard cubic feet) to Gg (giga-gram) for methane emissions. For the 1797141 MSCF/year, that is about 34.5 Gg/year, which is about 1/10 of our estimates of leakage. The results shows that the decrease rate reported by the gas company is larger than CLARS observations but the absolute leakage mass is significantly underestimated.

The mitigation actions taken by the government, including SoCalGas, has been summarized in SB 1371 and SB 1383, which are two bills by the California government to cut methane emissions. The bill SB1383 requires the state board to approve and begin implementing comprehensive strategy to reduce emissions of short-lived climate pollutants, and for methane to achieve a reduction by 40% below 2013 levels by 2030. The measures include adopting regulations to reduce methane emissions from livestock manure management operations and dairy manure management operations and to reduce the landfill disposal of organics. The bill 1371 specifically targets reducing natural gas leakage from the commission-regulated gas pipeline facilities that are intrastate transmission and distribution lines.

Information about SB1371 and SB1383:

[1] SB 1371 (Natural gas: leakage abatement) was approved by Governor on September 21, 2014: https://leginfo.legislature.ca.gov/faces/billTextClient.xhtml?bill_id=201320140SB1371

[2]SB 1383 (Short-lived climate pollutants: methane emissions: dairy and livestock: organic waste: landfills) was approved by Governor on September 19, 2016:

https://leginfo.legislature.ca.gov/faces/billTextClient.xhtml?bill_id=201520160SB1383

Reviewer #1 (Remarks to the Author):

I am satisfied with the extensive revisions done on the manuscript. The paper should be published.

Reviewer #2 (Remarks to the Author):

The authors were responsive to my earlier comments and I believe that the revised manuscript is improved and ready for publication.